# The Lateralization of Spatial Cognition in Table Tennis Players: Neuroplasticity in the Dominant Hemisphere

**DOI:** 10.3390/brainsci12121607

**Published:** 2022-11-23

**Authors:** Ziyi Peng, Lin Xu, Haiteng Wang, Tao Song, Yongcong Shao, Qingyuan Liu, Xiechuan Weng

**Affiliations:** 1School of Psychology, Beijing Sport University, Beijing 100084, China; 2State Key Laboratory of Cognitive Neuroscience and Learning, Beijing Normal University, Beijing 100875, China; 3IDG/McGovern Institute for Brain Research, Beijing Normal University, Beijing 100875, China; 4Department of Internal Medicine, Western Medical Branch of PLA General Hospital, Beijing 100144, China; 5Department of Neuroscience, Beijing Institute of Basic Medical Sciences, Beijing 100850, China

**Keywords:** spatial cognition, lateralization, neuroplasticity, table tennis player, event-related potentials

## Abstract

Spatial cognition facilitates the successful completion of specific cognitive tasks through lateral processing and neuroplasticity. Long-term training in table tennis induces neural processing efficiency in the visuospatial cognitive processing cortex of athletes. However, the lateralization characteristics and neural mechanisms of visual–spatial cognitive processing in table tennis players in non-sport domains are unclear. This study utilized event-related potentials to investigate differences in the spatial cognition abilities of regular college students (controls) and table tennis players. A total of 48 participants (28 controls; 20 s-level national table tennis players) completed spatial cognitive tasks while electroencephalography data were recorded. Task performance was better in the table tennis group than in the control group (reaction time: *P* < 0.001; correct number/sec: *P* = 0.043), P3 amplitude was greater in the table tennis group (*P* = 0.040), spatial cognition showed obvious lateralization characteristics (*P* < 0.001), table tennis players showed a more obvious right-hemisphere advantage, and the P3 amplitude in the right hemisphere was significantly greater in table tennis athletes than in the control group. (*P* = 0.044). Our findings demonstrate a right-hemisphere advantage in spatial cognition. Long-term training strengthened the visual–spatial processing ability of table tennis players, and this advantage effect was reflected in the neuroplasticity of the right hemisphere (the dominant hemisphere for spatial processing).

## 1. Introduction

Spatial cognition is a non-verbal information processing ability that allows one to understand and manipulate the environment. As such, it facilitates the representation, transformation, generation, and extraction of symbols. Extensive research has been conducted to explore the mechanisms underlying spatial cognition. McGee (1979) proposed that spatial cognitive ability includes two important components: spatial visualization ability and spatial orientation ability [1]. Linn and Petersen (1985) divided spatial ability into three components: spatial visualization ability, mental rotation, and spatial perception [2]. Spatial cognition includes visual analyses of shapes and contours as well as judgment and recognition of three-dimensional spatial relationships. Studies on spatial cognition often include visual and spatial tasks. Visual tasks mainly involve the recognition and re-recognition of objects, whereas spatial tasks mainly involve the orientation, reorientation, and three-dimensional operation of an object [2,3]. Spatial visualization abilities facilitate the rotation, manipulation, and distortion of stimulus objects in two or three dimensions in the mind. In addition, spatial working memory assists and regulates spatial visualization and orientation abilities and is considered the third most important component of spatial cognition. Spatial working memory facilitates task performance by storing limited visuospatial information under the control of the attentional center [4].

The cognitive attributes of spatial cognitive ability are completely different from those involved in speech. The functional asymmetry between the left and right hemispheres of the human brain is an important feature of human brain function [5]. Marc first proposed that the two cerebral hemispheres of the human brain are responsible for different cognitive behaviors and functions [6]. In terms of brain structure, differences in the network structure of each hemisphere are reflected in the asymmetry of gray matter and white matter structures [7]. The right hemisphere is known to be more efficiently organized and regionally interconnected than the left hemisphere. Anatomical differences reflecting hemispheric asymmetry are also observed in the connections between corresponding regions of the two hemispheres. Strong connectivity between the cerebral hemispheres is associated with the systematic co-activation of corresponding regions in cognitive tasks [8] and high correlations between corresponding brain regions with intrinsic functional connectivity [9]. Corresponding regions of the two hemispheres also exhibit a higher correlation in thickness than non-corresponding regions [10]. Changes in the relative strength of isotopic correlations in the brain may reflect varying degrees of lateralization. This anatomical asymmetry between the left and right hemispheres is established early in life and is influenced by genetic factors [11,12]. These structural differences reflect the specialization of the right hemisphere in broader processes (e.g., visuospatial integration tasks) and the leading role of the left hemisphere in particularly demanding specific processes (e.g., language and movement) [13]. Therefore, compared with the left hemisphere, the right hemisphere has a lateral advantage in spatial cognitive processing.

Gordon put forward the notion of the lateralization of cognitive function and suggested that this theory more effectively explained task performance and special skills than the theory of hemispheric dominance (e.g., visuospatial skills are primarily related to the right hemisphere, while verbal and sequential skills are primarily related to the left) [14]. Developmental studies on language and visuospatial function have demonstrated that, unlike anatomical asymmetry, functional lateralization develops during periods of relative maturity [15]. The maturation of lateralization is associated with improvements in visuospatial and linguistic abilities which are necessary for the development of efficient cognitive processes. Studies on the development of the right hemisphere and its functions have shown that in visual search and visuospatial memory tasks, the lateral processing of spatial cognitive brain functions increases with age [15,16]. Numerous neuropsychological studies suggest that the right hemisphere of the brain has an advantage in processing the spatial orientation of attention. The dominance of the right hemisphere emerges in the visual coding, processing, and synthesis of visual stimuli as well as stimulus configuration [17]. In a previous study on individuals completing spatial working memory tasks, we found that the P300 amplitude was significantly larger in the right hemisphere than in the left hemisphere [18]. P300, a component of ERP, can reflect deep processing in visual–spatial tasks. P300 (also termed P3) is affected by the objective probability of an event and the subjective expectation before the stimulus occurs. It is a sign of stimulus evaluation, attention capacity, processing time, and content update [19], reflecting the deployment of attention resources.

This right lateralization is known to be associated with the ventral frontoparietal attentional network [20,21], which intrinsically connects right asymmetry and the temporoparietal junction with the anterior insula. White matter connections between the lobes also provide anatomical evidence for this [22,23,24]. Moreover, visuospatial lateralization is also associated with the most ventral superior longitudinal tract and is related to the volumetric asymmetry of the right side.

Brain plasticity—also known as neuroplasticity—encompasses the vast changes that occur in all components of the central nervous system in response to internal and external stimuli throughout an individual’s life. These changes include functional and structural modifications that underlie individual development [25]. The main factors influencing brain plasticity include acquired learning, training, and experience. Although the brain is the source of behavior, the brain is also modified by the behavior it produces. The neural structure and corresponding functions of the brain undergo adaptive changes during learning and development [26]. Zatorre pointed out that the dynamic loop between the structure and function of the brain is the cognitive neural basis of learning and plasticity [27,28]. Adaptive changes in the brain (produced by acquired experience) anatomically manifest as plasticity in the structure of the cerebral cortex. Thus, it is widely accepted that the human brain has a plastic neural structure that helps it adapt to changes in the external environment [28].

Table tennis is a fast, net-separated, confrontational sport that requires athletes to cognitively process various types of spatial information—such as the direction, landing point, and rotation of the incoming ball—in a short period of time. Therefore, strong cognitive processing skills are required to play this sport. Long-term training in table tennis not only improves athletes’ performance but may also improve neuroplasticity in their visuospatial cognitive processing cortical regions, thus leading to high neural efficiency in the processing of visuospatial tasks.

Spatial cognition is associated with a moderate-to-high degree of plasticity [29]. Spatial cognitive processing ability is crucial for table tennis players because it directly affects the perception, processing, and judgment of fast and complex motion information. Since table tennis-specific sports and the general cognitive processing of spatial information involve a common neural processing pathway, the high neural efficiency of visuospatial processing is expected to exhibit plasticity. Cerebral cortical functions related to spatial cognition can be strengthened through long-term training exercises that are rapid, complex, and diverse in nature. In particular, the functional coupling between the optic nerve and motor circuit pathways is greatly affected by training. Movement is known to be highly correlated with information processing speed [30]. However, the lateralization characteristics and neural mechanisms of visuospatial cognitive processing in table tennis players engaged in non-sports tasks are still unclear and need further research.

In this study, we used event-related potentials (ERP) to explore the behavioral performance and neural activity of ordinary college students and table tennis players engaged in spatial cognitive processing tasks with different levels of difficulty. We tested whether table tennis players exhibited more obvious lateralization characteristics than ordinary college students when completing spatial cognitive processing tasks. In addition, we tested whether the P3 components induced by spatial cognitive tasks differ significantly between the left and right hemispheres.

## 2. Materials and Methods

### 2.1. Participants

A total of 50 participants were enrolled in the study (Table 1), including 28 male college students with no specialization (average age, 22.79 ± 2.27 years) and 22 table tennis players (average age, 20.00 ± 1.40 years). All table tennis players were second-level national athletes recruited from universities in Beijing. The athletes had trained for an average of 14.35 ± 2.29 h per week for 7.69 ± 4.79 years. The inclusion criteria were as follows: participants must (1) voluntarily participate in the study and be willing to provide written informed consent; (2) be physically healthy with normal uncorrected or corrected vision; (3) have no recent history of acute infection or symptoms of infection and no recent history of medication; (4) have good sleep habits, as indicated by the Pittsburgh Sleep Quality Index (score below 5); (5) have no history of mental or neurological disease and no symptoms of anxiety or depression; (6) have no history of smoking cigarettes or drinking coffee, alcohol, or tea; (7) be right-handed; (8) refrain from performing strenuous exercise before the testing.

### 2.2. Experimental Design

Perceptual priming—that is, the acceleration of the encoding process during object recognition—involves sensory-specific information insensitive to perceptual changes in non-diagnostic features. Perceptual priming allows one to identify specific sensory representations of objects. A core aspect of the model is the separation of perceptual priming and familiarity representations. For table tennis players, the shape of the ball and the location of the hitting point are familiar representations, and the two pieces of information are bundled to induce faster perceptual activation and faster responses. Spatial working memory is the ability to temporarily store a visuospatial memory of a certain size to complete a task under the control of the attentional center [4]. This modulates higher-level spatial abilities, such as mental rotation.

This study adopted a between-subject design. Stimulus materials were chosen based on the “Type Token” model proposed by Zimmer [31]. Considering the characteristics of table tennis, a circle with a notch (hereinafter referred to as a notch circle) was selected as the stimulus material. We used the 2-back spatial working memory paradigm combined with the special characteristics of table tennis to test the spatial cognitive ability of the participants.

### 2.3. Stimuli and Procedure

#### 2.3.1. The Notch Circle Task

The notch circle task (NCT) was based on the classical spatial 2-back working memory task. The stimulus material was a black notch circle. The notch angle was 15°, the notch position was calculated in the counterclockwise direction, and the center position corresponded to 45° (upper left). The task lasted for approximately 5 min and included 122 trials. During the experiment, a prompt message (a black “+” symbol) was presented in the center of the display for 200 ms. After the prompt message disappeared, a blank screen was displayed for 1 s. Following this, a circle was randomly presented on the screen (3 × 3 grid) for 400 ms. The inter-stimulus interval was 1600 ms. Participants were asked to match the current stimulus with the stimulus presented two trials earlier. If the spatial positions were consistent, the participants pressed the left mouse button with the index finger of the right hand; if the spatial positions were inconsistent, they pressed the right mouse button with the middle finger of the right hand. Consistent and inconsistent stimuli were presented in a random manner, and the ratio was maintained at 50% (Figure 1).

#### 2.3.2. The Rotating-Notch Circle Task

As in the NCT, the rotating-notch circle task (RNCT) was based on the classical spatial 2-back working memory task, and the stimulus material was a black notch circle. The notch angle was 15°, and there were four notch positions: the middle of the first notch corresponded to 45° (upper right), and those of the remaining notch positions corresponded to 135°, 225°, and 315° in the clockwise direction. The task lasted for approximately 5 min and included 122 trials. During the experiment, a prompt message (a black “+” symbol) was presented in the center of the display for 200 ms. A blank screen was displayed for 1 s, following which a randomly selected notch orientation was presented on the screen (3 × 3 grid) for 400 ms. The inter-stimulus interval was 1600 ms. Participants were asked to match the current stimulus with the stimulus presented two trials earlier and enter their responses as described above (see Section 2.3.1; Figure 2).

### 2.4. Electroencephalography Recordings

The EEG laboratory is dark, sound-proof, and electronically shielded. Referring to published research [18], the experimental program was compiled with E-Prime 2.0. EEG data were recorded using a 32-channel EEG recording and analysis system (NeuroScan, Charlotte, NC, USA), which has been extensively used (http://www.neuroscan.com/ accessed on 25 October 2022.) [32,33,34]. The installation electrode refers to the international 10–20 system of electrode placement. During the collection process, the horizontal and vertical electrooculograms of the subjects were recorded, and bilateral mastoids were used as reference electrodes. The EEG sampling frequency was 1000 Hz. By using electrode paste, the contact resistance between the electrode and scalp was reduced to below 5 kΩ. In the experiment, EEG data were saved for subsequent offline analyses.

### 2.5. Data Analysis

The ERP data were pre-processed with SCAN4.3 software. After the EEG preview, the ocular artifacts were removed by regression analysis. After filtering (band-pass was 0.05–30 Hz, 24 dB/oct), EEG segmentation (the ERP analysis window was 900 ms, and 100 ms before stimulation was used for baseline correction), baseline balancing, and artifact removal (±100 μV), the EEG-evoked potentials of all correct responses in the two tasks were averaged. The P3 (250–450 ms) components were analyzed for the following channels: F7, F3, FT7, FC3, C3, P3, F8, F4, FT8, FC4, CP4, and P4.

### 2.6. Statistical Analysis

A repeated-measures analysis of variance (ANOVA) was used to analyze the behavioral data and ERP findings using SPSS v.22 (IBM Corp., Armonk, NY, USA).

For the ERP analyses, we assessed the main and interaction effects of each group (controls and table tennis athletes), type of task (NCT and RNCT), brain hemisphere (left: average of the F7, F3, FT7, FC3, C3, and P3 channels; right: average of the F8, F4, FT8, FC4, C4, and P4 channels), and channel (F7, F8; F3, F4; FT7, FT8; FC3, FC4; C3, C4; P3, P4). The following behavioral parameters were also analyzed: mean reaction time (RT), accuracy, and the number of correct responses per unit time (= correct ratio*1000/correct time). Behavioral data were compared between the two groups (controls and table tennis athletes) and two types of tasks (NCT and RNCT).

The assumption of normality was verified with the Shapiro–Wilk test; the data demonstrated a normal distribution. We performed post hoc tests with the Greenhouse–Geisser correction for non-sphericity and the Bonferroni correction for multiple comparisons. The effect size (ES) was classified as small (0.01 < η^2^_P_ < 0.06), medium (0.06 < η^2^_P_ < 0.14), or large (η^2^_P_ > 0.14), and the significance level was set at *P* < 0.05. The results are presented as the mean ± the standard deviation. In addition, we performed a post hoc power analysis using G*power.

## 3. Results

### 3.1. Behavioral Performance

The data of two subjects were eliminated in post-processing due to excessive artifacts in the ERP data. Finally, the data of 28 controls and 20 table tennis players were included in the analysis.

The results of the behavioral experiments are presented in Table 2. The type of task had significant main effects on RT (*F* _(1, 46)_ = 18.585, *P* < 0.001, η^2^_P_ = 0.288, 1-β = 0.995), accuracy (*F* _(1, 46)_ = 52.185, *P* < 0.001, η^2^_P_ = 0.531, 1-β = 1), and the number of correct responses per unit time (*F* _(1, 46)_ = 18.469, *P* < 0.001, η^2^_P_ = 0.286, 1-β = 0.994). The results revealed that the RNCT was more difficult than the NCT. Group type had a significant main effect on the number of correct responses per unit time (*F* _(1, 46)_ = 4.350, *P* = 0.043, η^2^_P_ = 0.086, 1-β = 0.820), and table tennis athletes showed better performance in the tasks. The “group” factor did not have a significant main effect on accuracy (*F* _(1, 46)_ = 2.528, *P* = 0.119, η^2^_P_ = 0.052, 1-β = 0.760) or RT (*F* _(1, 46)_ = 1.719, *P* = 0.196, η^2^_P_ = 0.036, 1-β = 0.720), and the results indicated a trend of better task performance among table tennis athletes (Figure 3). No other main or interaction effects were statistically significant.

### 3.2. Amplitude

The amplitudes at the F7, F3, FT7, FC3, C3, P3, F8, F4, FT8, FC4, CP4, and P4 electrodes during the two tasks are summarized in Table 3.

The type of task had a significant main effect on P3 amplitude (*F* _(1, 46)_ = 4.314, *P* = 0.043, η^2^_P_ = 0.086, 1-β = 0.820), and the P3 amplitude was larger for the NCT than for the RNCT. Group type had a significant main effect on P3 amplitude (*F* _(1, 46)_ = 4.451, *P* = 0.040, η^2^_P_ = 0.088, 1-β = 0.821), and the P3 amplitude was larger for table tennis athletes than for the control group. The hemisphere also had a significant main effect on P3 amplitude (*F* _(1, 46)_ = 23.546, *P* < 0.001, η^2^_P_ = 0.339, 1-β = 0.999), and the P3 amplitude was larger in the right hemisphere than in the left hemisphere (Figure 4 and Figure 5).

Group type and hemisphere did not have a significant interaction effect on P3 amplitude (*F* _(1, 46)_ = 0.148, *P* = 0.702, η^2^_P_ = 0.003, 1-β = 0.740). However, a simple effect analysis revealed that the P3 amplitude in the right hemisphere was significantly larger among table tennis athletes than in the control group (*P* = 0.044, η^2^_P_ = 0.085), although there was no significant between-group difference in the P3 amplitude in the left hemisphere. In addition, the P3 amplitude was largest (*F* _(5, 230)_ = 36.676, *P* < 0.001, η^2^_P_ = 0.444, 1-β = 0.999) in the parietal–occipital lobe (CP3, CP4, P3, and P4).

### 3.3. Latency

The latencies at the F7, F3, FT7, FC3, C3, P3, F8, F4, FT8, FC4, CP4, and P4 electrodes during the two tasks are summarized in Table 4.

The type of task did not have a significant main effect on P3 latency (*F* _(1, 46)_ = 0.279, *P* = 0.600, η^2^_P_ = 0.006, 1-β = 0.693), and P3 latency was not significantly different between the NCT and RNCT. Group type did not have a significant main effect on P3 latency (*F* _(1, 46)_ = 0.214, *P* = 0.646, η^2^_P_ = 0.005, 1-β = 0.717), and P3 latency was not significantly different between the control and table tennis groups. The brain hemisphere had a significant main effect on P3 latency (*F* _(1, 46)_ = 2.664, *P* = 0.109, η^2^_P_ = 0.055, 1-β = 0.767); however, P3 latency was not significantly different between the left and right hemispheres (Figure 3 and Figure 4).

Group type and hemisphere did not have a significant interaction effect on P3 latency (*F* _(1, 46)_ = 0.957, *P* = 0.333, η^2^_P_ = 0.020, 1-β = 0.673). However, P3 latency was highest (*F* _(5, 230)_ = 18.028, *P* < 0.001, η^2^_P_ = 0.282, 1-β = 0.881) in the parietal–occipital lobe (CP3, CP4, P3, and P4).

## 4. Discussion

In this study, we explored the behavioral performance and the characteristics of neural activity related to spatial cognitive processing among ordinary college students and table tennis athletes. The performance of all participants worsened with increasing task difficulty as follows: reaction time increased and accuracy decreased. However, overall task performance was better among table tennis athletes than in the control group. The table tennis group exhibited improved spatial cognitive processing compared with the control group in more difficult tasks with high cognitive loads [27]. In addition, the number of correct responses per unit time was significantly higher in the table tennis group than in the control group. When completing tasks, the participants may have adopted a response strategy that sacrificed the rate of correct responses in favor of shortening the reaction time. However, the reaction time of the table tennis group was slightly faster than that of the control group. This difference may be due to the table tennis group choosing quicker reaction strategies or having stronger abilities related to attention and inhibition control [35].

The table tennis players and controls showed different neural processing characteristics in spatial cognitive tasks. The P3 amplitude increased with task difficulty and was larger in table tennis players than in the controls. The P3 component reflects the deployment of attentional resources, and the P3 amplitude reflects an individual’s ability to distinguish and evaluate target stimuli [19]. Long-term special training can help table tennis players actively adapt to fast, complex, and changeable situations. However, when ordinary college students completed the spatial cognitive tasks, the greater task difficulty affected their subjective assessment of cognitive task competency; therefore, these students experienced reduced resource inputs from the brain [36]. Previous research on athletes faced with complex or difficult tasks has revealed an inconsistent correlation between performance and the degree of activation of the cognitive cortex or the degree of neural resource consumption. This finding indicates that task complexity or difficulty regulates the activation of the cerebral cortex. Researchers have also explored how the complexity or difficulty of tasks affects the relationship between performance and activation [37,38,39,40]. The results showed that when faced with low- or medium-difficulty tasks, the cortical activation of both low-performance and high-performance individuals gradually increased with increasing task complexity/difficulty, although cortical activation increased to a lower degree among high-performance individuals. However, as the complexity or difficulty of the task exceeded a certain threshold, high-performing individuals demonstrated better task performance by further activating their cortical neural resources. In contrast, low performers reached their cognitive load limit due to their limited neurocognitive resources. That is, low-performing individuals cannot successfully complete more complex or difficult tasks even if they use all their neural resources, resulting in reduced performance accompanied by constant or reduced cortical activation.

Based on the theory of brain plasticity, some scholars have put forward the neural efficiency hypothesis, which proposes that individuals with greater cognitive abilities can use fewer neural resources to achieve higher cognitive levels [41]. However, the current research reports on neural efficiency are inconsistent [42,43], especially for challenging cognitive tasks (even if neural efficiency is somewhat transferable) [44,45]. Individuals who exhibit high cognitive abilities show greater activation in the task-related frontoparietal cortex than individuals who exhibit low cognitive abilities. In the field of sports, increasingly, more studies have begun to explore the neural efficiency of athletes who have undergone long-term professional training in specific sports. For example, compared to novice/non-athletes, karate athletes exhibit lower event-related desynchronization in the alpha-band of the mirror system when viewing pictures of real competitions [46]. There does not seem to be a simple linear relationship between an athlete’s sport proficiency and ERP amplitude. Among athletes engaged in inhibitory tasks [47], the increase in P300 amplitude may be related to the mobilization of early resources, the conservation of late conscious resources, and the inhibition of attention to stimuli unrelated to the sports process (in the sports context). This effect improves athletes’ performance and helps them actively adapt to rapidly changing sports situations [48,49]. In other words, the high neural efficiency of athletes is reflected in the reduction in resource input during pre-processing, which ensures that there are enough resources for late-stage advanced cognitive control. However, some studies on working memory have also shown that the P300 amplitude decreases as task complexity or difficulty increases [50,51].

One of the most important findings of this study is that spatial cognitive processing shows obvious lateralization with right-dominant brain processing. Table tennis players showed a more obvious right-hemisphere advantage, and the P3 amplitude in the right hemisphere was significantly greater in table tennis athletes than in the control group (Figure 3 and Figure 4). Moreover, there was no significant between-group difference in the P3 amplitude of the left hemisphere. When processing spatial information, the right hemisphere uses more cognitive resources. Neuropsychological studies have shown that the spatial location of attention has a right-hemisphere advantage. Lateral spatial neglect mostly occurs in the right hemisphere of the brain [52,53]. Neuroimaging evidence shows that the dorsal parietal lobe participates in the storage of spatial working memory [54], and the activation intensity of the dorsolateral prefrontal cortex increases with an increase in working memory load [55,56]. Our results also showed that the activities of the parietal–occipital region were more obvious when the participants completed the task. The right hemisphere is the dominant hemisphere for visual–spatial processing, and our results show that in both groups, the spatial task-induced P3 amplitude was greater in the right hemisphere than in the left hemisphere. Spatial intelligence is an important part of multiple intelligence, and the neural plasticity of the brain emphasizes the importance of acquired experience [57]. Table tennis players engaging in sports-related spatial cognitive tasks can use their brains’ neural resources in highly efficient ways, as reflected in the activation level of the frontal and parietal cortex and changes in the rhythmic coupling of various brain regions (including the frontal-occipital, frontal-temporal, and frontal-parietal regions). This efficiency is particularly reflected in the late processing period [58]. These findings are consistent with the trends in the late P3 component in this study.

The athlete’s brain is a good model for exploring neural plasticity because most athletes engage in long-term training and practice from early childhood [59,60]. Through long-term professional training, they demonstrate outstanding abilities for rapid stimulus discrimination, decision making, and attention deployment. Several studies have reported plastic changes in the resting-state functional connectivity of professional athletes [61,62,63,64]. Athletes trained in highly reactive sports (e.g., table tennis) show highly efficient visual movement, which is characterized by automaticity, speed, and accuracy [60]. This efficiency is achieved through long-term training, which benefits from highly developed visual attention skills and sports strategies. In addition to exploring the plasticity of resting-state functional networks, some researchers also explored the influence of long-term training on brain activation in task states [65]. The adaptive enhancement of sport-specific processing is the basis for excellent athletes to achieve outstanding results [66]. However, long-term, domain-specific training has compound effects on general perception and the cognitive domain [67,68,69]. Based on our findings and evidence from anatomical and functional studies, we speculate that table tennis players may exhibit an advantage in spatial cognitive processing due to long-term training to strengthen the visual-spatial processing ability of the right hemisphere [67].

Our study had some limitations. First, we only used male volunteers; therefore, our conclusions may not be applicable to women. When performing spatial memory tasks, the left hemisphere is more active in women, whereas the right hemisphere is more active in men. Moreover, men use integrated strategies to a greater degree to remember spatial locations [70]. Future studies may include female subjects to evaluate the relevant differences. Second, ERP data have limited spatial resolution and cannot precisely provide the characteristics of neuronal or cluster activities with high spatial resolution. Third, our low sample size may have affected our ability to generate reliable results. If more high-level athletes are included in future research, the results will be more stable. Fourth, because talented people are more likely to be included in a sports team when athletes are selected, a cross-sectional study cannot fully prove that the cognitive advantage of table tennis is 100% caused by the acquired training. In the future, longitudinal research may be adopted to explore the cognitive effects of exercise for ordinary people.

## 5. Conclusions

This study used ERP technology to explore the differences in spatial cognitive processing and lateralization between table tennis players and ordinary college students. Spatial cognitive processing had a right-hemisphere advantage, and the P3 amplitude was significantly greater in the right hemisphere than in the left hemisphere when completing spatial tasks. Compared with ordinary college students, table tennis players showed advantages in spatial cognitive processing and task performance, and the P3 amplitude was greater among table tennis payers than in the control group. This difference was more obvious in the right hemisphere. Long-term training strengthened the visual–spatial processing ability of table tennis players, and this advantage effect was reflected in the neuroplasticity of the right hemisphere (the dominant hemisphere for spatial processing).

## Figures and Tables

**Figure 1 brainsci-12-01607-f001:**
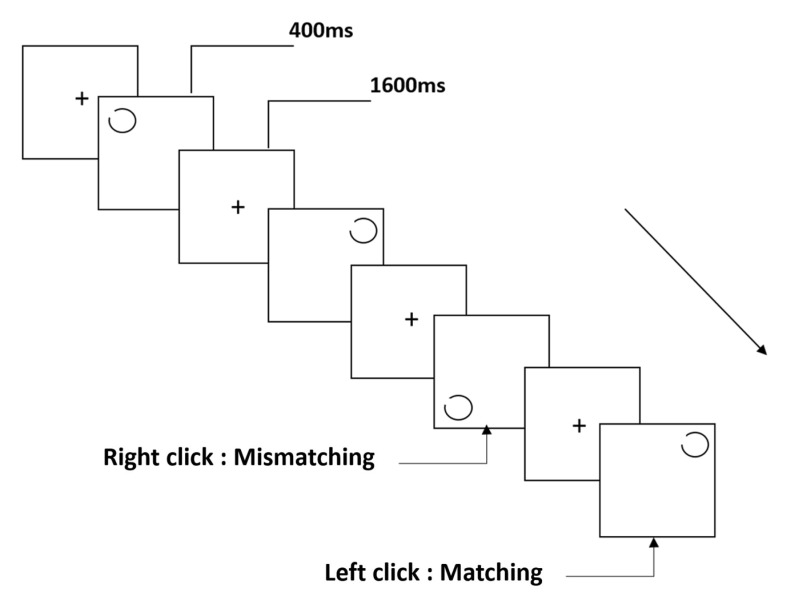
Schematic diagram of the notch circle task.

**Figure 2 brainsci-12-01607-f002:**
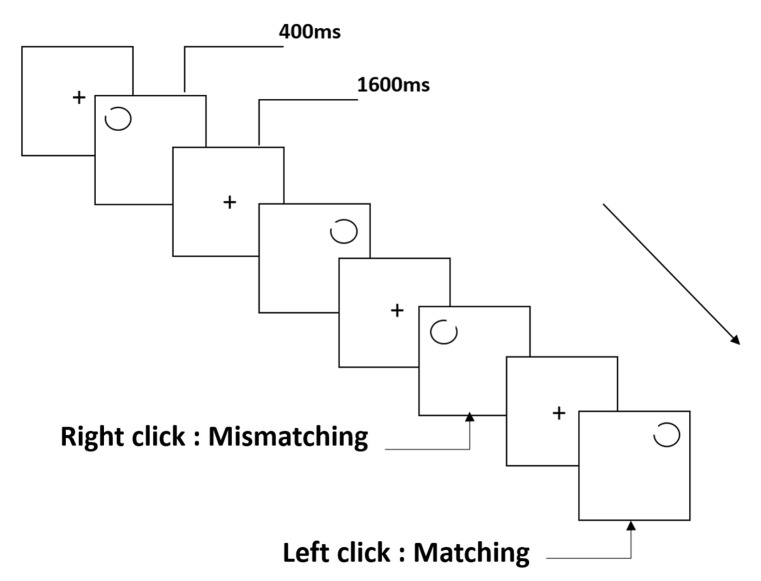
Schematic diagram of the rotating-notch circle task.

**Figure 3 brainsci-12-01607-f003:**
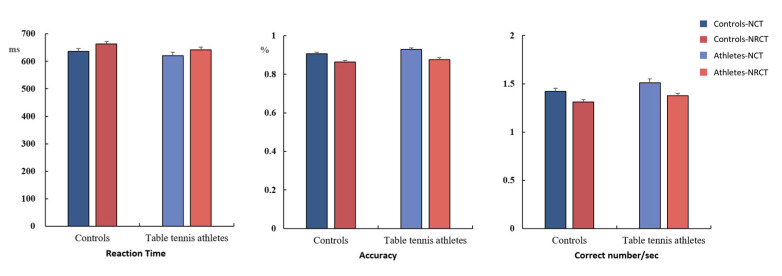
Reaction time, accuracy, and number of correct responses per unit time (mean ± standard deviation). NCT: notch circle task; RNCT: rotating-notch circle task.

**Figure 4 brainsci-12-01607-f004:**
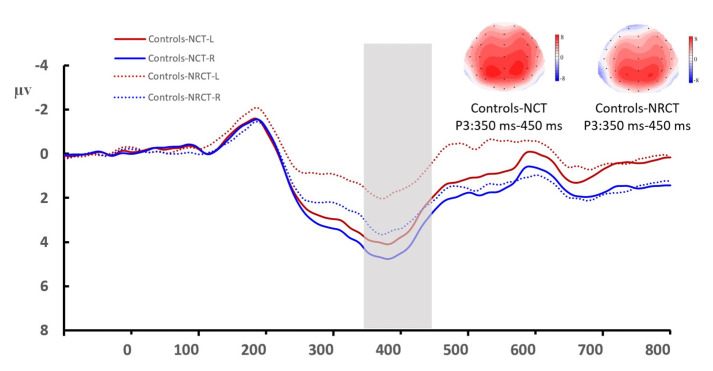
Mean amplitude of the P3 component in the two hemispheres in the control group. Averaged data of the left hemisphere are from the F3, F7, FT3, FT7, CP3, and P3 electrodes. Averaged data of the right hemisphere are from the F4, F8, FT4, FT8, CP4, and P4 electrodes. The topographies correspond to average activity in the time window of 350–450 ms (indicated by the gray bar) around the local peaks. NCT: notch circle task; RNCT: rotating-notch circle task.

**Figure 5 brainsci-12-01607-f005:**
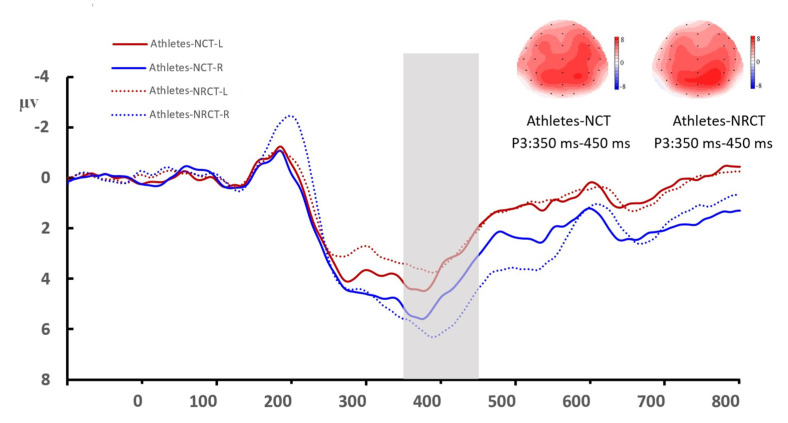
Mean amplitude of the P3 component in the two hemispheres among table tennis athletes. Averaged data of the left hemisphere are from the F3, F7, FT3, FT7, CP3, and P3 electrodes. Averaged data of the right hemisphere are from the F4, F8, FT4, FT8, CP4, and P4 electrodes. The topographies correspond to average activity in the time windows of 350–450 ms (indicated by the gray bar) around the local peaks. NCT: notch circle task; RNCT: rotating-notch circle task.

**Table 1 brainsci-12-01607-t001:** Demographic information of each group (mean ± standard deviation).

Group	Controls	Table Tennis Athletes
Number	28	22
Gender	Male	Male
Age (y)	22.79 ± 2.27	20.00 ± 1.40
Mass (kg)	62.81 ± 4.16	71.69 ± 1.15 *
Height (cm)	176.32 ± 5.40	177.2 ± 6.35
BMI (kg·m^−2^)	20.20 ± 1.15	22.83 ± 1.65 *
Experience (y)	-	7.69 ± 4.79

BMI = body mass index, * = statistically significant with *p* < 0.05.

**Table 2 brainsci-12-01607-t002:** Behavioral performance (mean ± standard deviation) in the two types of tasks in the control and table tennis groups.

Behavior Indictors	Group	Notch Circle Task (*M* ± SD)	Rotating-Notch Circle Task (*M* ± SD)
Accuracy	Controls	0.91 ± 0.04	0.86 ±0.05
	Table tennis athletes	0.93 ± 0.04	0.88 ±0.04
Reaction Time	Controls	635.41 ± 64.10	662.45 ± 51.48
	Table tennis athletes	620.28 ± 42.15	641.53 ± 32.69
Correct number/sec	Controls	1.42 ± 0.21	1.31 ± 0.13
	Table tennis athletes	1.51 ± 0.14	1.37 ± 0.12

**Table 3 brainsci-12-01607-t003:** Grand-average peak amplitude (μV) of the P3 component for correct responses across multiple electrode sites in the control and table tennis groups.

	Notch Circle Task (*M* ± SD)		Rotating-Notch Circle Task (*M* ± SD)	
	Controls	Table Tennis Athletes	Controls	Table Tennis Athletes
F7	3.78 ± 3.29	5.14 ± 3.70	2.30 ± 3.72	4.09 ± 3.53
F8	5.38 ± 2.70	6.49 ± 3.04	4.00 ± 2.89	6.24 ± 2.94
F3	6.02 ± 3.00	6.50 ± 3.65	4.01 ± 2.91	5.87 ± 3.70
F4	6.89 ± 3.03	7.33 ± 3.14	5.58 ± 2.07	7.05 ± 3.30
FT7	4.56 ± 3.28	5.26 ± 3.55	2.43 ± 3.06	4.33 ± 3.10
FT8	5.52 ± 2.16	6.32 ± 2.96	4.01 ± 2.77	5.91 ± 2.82
FC3	6.68 ± 2.56	7.05 ± 3.17	5.35 ± 2.44	6.93 ± 4.37
FC4	7.61 ± 3.50	7.69 ± 3.49	6.34 ± 2.23	7.71 ± 3.62
CP3	5.74± 1.44	6.46 ± 2.44	5.74 ± 1.93	6.85 ± 3.43
CP4	6.38± 1.83	7.69 ± 2.73	6.48 ± 2.09	8.63 ± 3.81
P3	5.44± 1.56	6.24 ± 2.22	5.64 ± 2.01	6.86 ± 3.05
P4	5.69± 1.85	6.85 ± 2.64	5.97 ± 1.72	8.29 ± 3.62

**Table 4 brainsci-12-01607-t004:** Grand-average peak latency (ms) of the P3 component for correct responses across multiple electrode sites in the control and table tennis groups.

	Notch Circle Task (*M* ± SD)		Rotating-Notch Circle Task (*M* ± SD)	
	Controls	Table Tennis Athletes	Controls	Table Tennis Athletes
F7	344.27 ± 40.96	335.45 ± 32.00	342.32 ± 40.08	328.26 ± 40.71
F8	365.42 ± 44.42	344.74 ± 51.51	330.25 ± 32.50	345.89 ± 52.84
F3	336.42 ± 47.22	322.84 ± 40.51	321.95 ± 41.81	332.39 ± 37.53
F4	350.56 ± 41.68	329.66 ± 46.63	318.64 ± 41.09	335.47 ± 37.01
FT7	359.27 ± 43.99	342.16 ± 28.52	351.17 ± 39.83	332.26 ± 45.99
FT8	365.83 ± 45.61	349.82 ± 49.63	351.39 ± 39.52	352.79 ± 51.11
FC3	346.79 ± 45.37	330.66 ± 47.07	334.32 ± 38.85	335.39 ± 43.54
FC4	346.69 ± 41.99	330.50 ± 42.90	326.77 ± 40.98	338.26 ± 38.51
CP3	352.96 ± 38.69	348.29 ± 37.60	354.58 ± 29.77	357.76 ± 37.57
CP4	355.73 ± 42.87	357.84 ± 33.22	362.06 ± 32.52	357.53 ± 37.95
P3	350.46 ± 38.60	351.74 ± 37.23	362.42 ± 37.15	365.45 ± 38.92
P4	353.92 ± 44.01	358.21 ± 34.89	358.35 ± 38.39	365.53 ± 38.27

## Data Availability

The datasets generated for this study are available on request to corresponding authors.

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
