# Peer review of "The Lateralization of Spatial Cognition in Table Tennis Players: Neuroplasticity in the Dominant Hemisphere"

_brainsci, 2022, doi:10.3390/brainsci12121607_

Round 1
Reviewer 1 Report
Title: The lateralization of spatial cognition in table tennis players: neuroplasticity in the dominant hemisphere
This article seems well built and brings evidence of a phenomenon not yet fully understood and that certainly deserves further study.
Some points of revision are provided below
Abstract
- I suggest including the data with p values
Materials and Methods
- The authors need to include anthropometric data (body mass / body height/ BMI), training experience – weekly training (h) – level according to national/international ranking for Table Tennis players, to better replicate this study
- The experimental design needs to be improved, thus…. Unclear how the authors tested all participants (setting, I suggest a figure to be better representative, for example:
- https://pubmed.ncbi.nlm.nih.gov/30765917/
- https://pubmed.ncbi.nlm.nih.gov/31009428/
- the device used should be validated e reliable (repeatability), accuracy and precision need to be included….now all this information’s are missing
Statistical analysis
- The significance level – the sample power/size – effect size – Anova – Fisher – eta square……are missing
Figure 4-5: y axes is missing…. About the u.m.
After this major improvement I can update a revision
Reviewer 2 Report
Thank you for the opportunity to review this study. I would like to congratulate the authors for a good work and a well written paper. Here are some comments that can be used to improve the manuscript.
1. General comment: I think the term »neuroplasticity« is used wrong in some cases. E.g., »table tennis induces neuroplasticity«, which would suggest that neuroplasticity is a change itself (but this is not the case, when it comes to the definitions, neuroplasticity is the ability to changed/modulated, not the change itself).
2. Abstract is generally well written, but I miss some concrete data (descriptive and/or effect sizes with p-values)
3. I looked up this reference as I was not familiar with it “Gordon put forward the notion of the lateralization of cognitive function and suggested that this theory more effectively explained homework performance and special skills than the theory of hemispheric dominance [14].” – I think it would be good to add the population in question at the end of the sentence, and also explain what test were used – “homework performance” is a bit vague
4. When mentioning P300 amplitude in the intro for the first time, I would reshape the sentence to make it clear that this is an EEG-based parameter (just for the sake of readers that are less familiar with neurophysiology)
5. Please include some more data for table tennis players, beyond saying they are “second-level” (it quite a vague term – are they professional players or not?) I would include at least the years of playing table tennis and number of trainings per week, if possible.
6. Although you excluded participants in case of alcohol/coffee/tea consumption, were there any other criteria that they have to meet or follow – for instance, not to perform strenuous exercise before the testing?
7. Did you check the normality of the data distribution?
8. Discussion is very well written. One additional thought on the limitations (also related to statement that athlete are a good model to use) – using cross-sectional design, it is impossible to say with 100 % confidence if the table tennis training cause the changes in spatial cognitive ability/lateralization, or if the individuals having these traits to begin with are more likely to become successful tennis table players.
Round 2
Reviewer 1 Report
also if the authors provided new informations ....improving the article.....several data are missing as suggested on my previously revision:
- body mass/ height
- training experience as weekly training
- precision / accuracy about the device used
- in results section are missing sample power - effect size
abstract is incomplete and unclear
Reviewer 2 Report
All comments have been addressed.
Author Response
Thank you for your letter and for the reviewers’ comments concerning our manuscript entitled “The lateralization of spatial cognition in table tennis players: neuroplasticity in the dominant hemisphere”. All of the comments were valuable and helpful for revising and improving our paper.